# Basis Function Encoding of Numerical Features in Factorization Machines for Improved Accuracy

## Abstract

Factorization machine (FM) variants are widely used for large scale real-time content recommendation systems, since they offer an excellent balance between model accuracy and low computational costs for training and inference. These systems are trained on tabular data with both numerical and categorical columns. Incorporating numerical columns poses a challenge, and they are typically incorporated using a scalar transformation or binning, which can be either learned or chosen a-priori. In this work, we provide a systematic and theoretically-justified way to incorporate numerical features into FM variants by encoding them into a vector of function values for a set of functions of one's choice.

We view factorization machines as approximators of *segmentized* functions, namely, functions from a field's value to the real numbers, assuming the remaining fields are assigned some given constants, which we refer to as the segment. From this perspective, we show that our technique yields a model that learns segmentized functions of the numerical feature spanned by the set of functions of one's choice, namely, the spanning coefficients vary between segments. Hence, to improve model accuracy we advocate the use of functions known to have strong approximation power, and offer the B-Spline basis due to its well-known approximation power, availability in software libraries, and efficiency. Our technique preserves fast training and inference, and requires only a small modification of the computational graph of an FM model. Therefore, it is easy to incorporate into an existing system to improve its performance. Finally, we back our claims with a set of experiments that include a synthetic experiment, performance evaluation on several data-sets, and an A/B test on a real online advertising system which shows improved performance. The results can be reproduced with the code in the supplemental material.

## 1 Introduction

Traditionally, online content recommendation systems rely on predictive models to choose the set of items to display by predicting the affinity of the user towards a set of candidate items. These models are trained on feedback gathered from a log of interactions between users and items from the recent past. For systems such as online ad recommenders with billions of daily interactions, speed is crucial. The training process must be fast to keep up with changing user preferences and quickly deploy a fresh model. Similarly, model inference, which amounts to computing a score for each item, must be rapid to select a few items to display out of a vast pool of candidate items, all within a few milliseconds. Factorization machine (FM) variants, such as Rendle (2010); Juan et al. (2016); Pan et al. (2018); Sun et al. (2021), are widely used in these systems due to their ability to train incrementally, and strike a good balance between being able to produce accurate predictions, while facilitating fast training and inference.

The training data consists of past interactions between users and items, and is typically given in tabular form, where the table's columns, or *fields*, have either categorical or numerical features. For example, "gender" or "time since last visit" are fields, whereas "Male" and "10 hours" are corresponding features. In recommendation systems that rely on FM variants, each row in the table is typically encoded as a concatenation of field encoding vectors. Categorical fields are usually one-hot encoded, whereas numerical fields are conventionally binned to a finite set of intervals to form a

categorical field, and one-hot encoding is subsequently applied. A large number of works are devoted to the choice of the intervals, e.g. Dougherty et al. (1995); Peng et al. (2009); Liu et al. (2002); Gama & Pinto (2006). Regardless of the choice, the model's output is a *step function* of the value of a given numerical field, assuming the remaining fields are kept constant, since the same interval is chosen independently of where the value falls in a given interval. For example, consider a model training on a data-set with age, device type, and time the user spent on our site. For the segment of 25-years old users using an iPhone the model will learn some step function, whereas for the segment of 37-years old users using a laptop the model may learn a (possibly) different step function.

However, the optimal segmentized functions the model aims to learn, that describe the user behavior, aren't necessarily step functions. Typically, such functions are continuous or even smooth, and there is a gap between the approximating step functions the model learns, and the optimal ones. In theory, a potential solution is simply to increase the number of bins. This increases the approximation power of step functions, and given an infinite amount of data, would indeed help. However, the data is finite, and this can lead to a *sparsity* issue - as the number of learning samples assigned to each bin diminishes, it becomes increasingly challenging to learn a good representation of each bin, even with large data-sets, especially because we need to represent all segments simultaneously. This situation can lead to a degradation in the model's performance despite having increased the theoretical approximation power of the model. Therefore, there is a limit to the accuracy we can achieve with binning on a given data-set. See Figure 1.

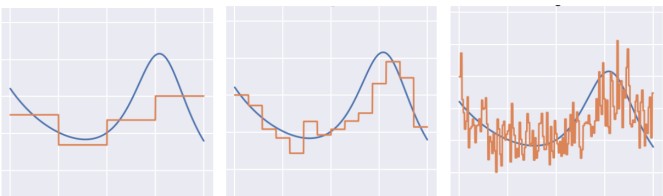

Figure 1: For a given segment, learned segmentized step function approximations (orange) of a true function (blue) which was used to generate a synthetic data-set. In the left - too few bins, bad approximation. In the middle - a balanced number of bins, moderate approximation. On the right - many bins, but approximation gets even worse due to a sparsity issue.

In this work, we propose a technique to improve the accuracy of FM variants by reducing the approximation gap *without* sparsity issues, while preserving training and inference speeds. Our technique is composed of encoding a numerical field using a vector of functions, which we refer to as *basis functions*, and a minor modification to the computational graph of an FM variant. The idea of using basis functions is of course a standard practice with linear models, but combined with FMs it is surprisingly powerful, and radically differs from the same technique applied to linear models.

Indeed, we present an elementary lemma showing that the resulting model learns a *segmentized* output function spanned by chosen basis, meaning that spanning coefficients depend on the values of the remaining fields. This is, of course, an essential property for recommendation systems, since indeed users with different demographic or contextual properties may behave differently. To the best of our knowledge, the idea of using arbitrary basis functions with FMs and the insights we present regarding the representation power of such a combination are new.

Based on the generic observation above, we offer the B-Spline basis (de Boor, 2001, pp 87) defined on the unit interval on uniformly spaced break-points, composed onto a transformation that maps the feature to the unit interval. The number of break-points (knots) is a hyper-parameter. The strong approximation power of splines ensures that we do not need a large number of break-points, meaning that we can closely approximate the optimal segmentized functions, without introducing sparsity issues. Moreover, to make integration of our idea easier in a practical production-grade recommendation system, we present a technique to seamlessly integrate a model trained using our proposed scheme into an existing recommendation system that employs binning, albeit with a controllable reduction in accuracy. Although a significant part of this work considers the B-Spline basis, the techniques we present can be used with an arbitrary basis.

To summarize, the main contributions of this work are: (a) *Basis encoding* We propose encoding numerical features using a vector of basis functions, and introducing a minor change to the

computational graph of FM variants. (b) *Spanning properties* We show that using our method, any model from a family that includes many popular FM variants, learns a *segmentized* function spanned by the basis of our choice, and inherits the approximation power of that basis. (c) *B-Spline basis* We justify the use of the B-Spline basis from both theoretical and practical aspects, and demonstrate their benefits via numerical evaluation. (d) *Ease of integration into an existing system* We show how to integrate a model trained by our method into an existing recommender system which currently employs numerical feature binning, to significantly reduce integration costs. (e) *Simplified numerical feature engineering* the strong approximation power of the cubic B-Spline basis allows building accurate models without investing time and effort to manually tune bin boundaries, and use simple uniform break-points instead.

## 1.1 RELATED WORK

If we allow ourselves to drift away from the FM family, we can find a variety of papers on neural networks for tabular data (Arik & Pfister, 2021; Badirli et al., 2020; Gorishniy et al., 2021; Huang et al., 2020; Popov et al., 2020; Somepalli et al., 2022; Song et al., 2019; Hollmann et al., 2022). Neural networks have the potential to achieve high accuracy and can be incrementally trained on newly arriving data using transfer learning techniques. Additionally, due to the *universal approximation* theorem (Hornik et al., 1989), neural networks are capable of representing a segmentized function from any numerical feature to a real number. However, the time required to train and make inferences using neural networks is significantly greater than that required for factorization machines. Even though some work has been done to alleviate this gap for neural networks by using various embedding techniques (Gorishniy et al., 2022), they have not been able to outperform other model types. As a result, in various practical applications, FMs are preferred over NNs. For this reason, in this work we focus on FMs in our work, and specifically on decreasing a gap between the representation power of FMs and NNs without introducing a significant computational and conceptual complexity.

A very simple but related approach to ours was presented in Covington et al. (2016). The work uses neural networks and represents a numerical value $z$ as the triplet $(z, z^2, \sqrt{z})$ in the input layer, which can be seen as a variant of our approach using a basis of three functions. Another approach which bears similarity to ours also comes from works which use deep learning (Cheng, 2022; Gorishniy et al., 2021; Song et al., 2019; Guo et al., 2021). In these works, first-order splines are used in the input layer to represent continuity, and the representation power of a neural network compensates for the weak approximation power of first-order splines. Here we do the opposite - we use the stronger approximation power of cubic splines to compensate for the weaker representation power of FMs.

The recent work of David (2022) uses a technique which appears to resemble ours, but takes the perspective of function approximation on a bounded domain, applies to regular FM models only, and expands towards higher orders of interaction. In contrast, our work takes the recommender system perspective and has a significantly broader theoretical and practical scope: it applies to a wide range of FM variants, includes handling of unbounded domains and interaction with categorical features, and reports A/B test results on a real-world ad recommendation system.

Finally, a work on tabular data cannot go without mentioning *gradient boosted decision trees* (GBDT) (Chen & Guestrin, 2016; Ke et al., 2017; Prokhorenkova et al., 2018), which are known to achieve state of the art results (Gorishniy et al., 2021; Shwartz-Ziv & Armon, 2022). However, GBDT models aren't useful in a variety of practical applications, primarily due to significantly slower inference speeds, namely, it is challenging and costly to use GBDT models to rank hundreds of thousands of items in a matter of milliseconds.

## 2 FORMAL PROBLEM STATEMENT

We consider tabular data-sets with $m$ fields, each can be either numerical or categorical. A row in the data-set $(z_1, \ldots z_m)$, is encoded into a feature vector:

$$\boldsymbol{x} = \mathbf{enc}(z_1, \ldots z_m) \equiv \begin{bmatrix} \mathbf{enc}_1(z_1) \\ \vdots \\ \mathbf{enc}_m(z_m) \end{bmatrix},$$

where the $\mathbf{enc}_f$ function encodes field $f$ into a vector, e.g. one-hot encoding of a single-valued categorical field. The vector $\boldsymbol{x}$ is then fed into a model. With a slight abuse of notation, we will denote by $\phi_f$ the model's segmentized output, which is the output as a function of the value of a field $f$, assuming the remaining fields are some given constants. As in any supervised learning task, $\phi_f$ aims to approximate some unknown optimal segmentized function.

In the vast majority of cases, a numerical field $f$ is either encoded using a scalar transform $\mathbf{enc}_f : \mathbb{R} \to \mathbb{R}$, or using binning - the numerical domain is partitioned into a finite set of intervals $\mathbf{enc}_f(z_f)$ is a one-hot encoding of the interval $z_f$ belongs to. Since for any $z_f \in [a_i, a_{i+1})$ we have the same encoding, a deterministic model will produce the same output, and therefore $\phi_f$ is a step function.

Step functions are weak approximators in the sense that many cut-points are required to achieve good approximation accuracy. For example, Lipschitz-continuous functions can be approximated by a step function on $\ell$ intervals up to an error of only $O(\frac{1}{\ell})$ (de Boor, 2001, pp 149). Hence, with binning we need to strike an intricate balance between the theoretically-achievable approximation accuracy and our ability to achieve it because of sparsity issues, as illustrated in Figure 1. Note, that the above observation does not depend on weather the bins are chosen a-priori or learned. In this work we propose an alternative to binning, by encoding numerical features in a way that allows achieving more of the theoretical accuracy before sparsity issues take effect.

## 2.1 THE FACTORIZATION MACHINE FAMILY

In this work we consider several model variants, which we refer to as the factorization machine family. The family includes celebrated factorization machine (FM) (Rendle, 2010), the field-aware factorization machine (FFM) (Juan et al., 2016), the field-weighted factorization machine (FwFM) (Pan et al., 2018), and the field-matrixed factorization machine (FmFM) (Sun et al., 2021; Pande, 2021), that generalized the former variants.

As a convention, we denote by $f_i$ the field which was used to encode the $i^{\text{th}}$ component of the feature vector $\boldsymbol{x}$, and denote arbitrary fields by $f, e$. We also denote by $\langle \boldsymbol{x}, \boldsymbol{y} \rangle_{\boldsymbol{P}} = \boldsymbol{x}^T \boldsymbol{P} \boldsymbol{y}$ the "inner product"[1] associated with some matrix $\boldsymbol{P}$. The FmFM model computes

$$\phi_{\text{FmFM}}(\boldsymbol{x}) = w_0 + \langle \boldsymbol{x}, \boldsymbol{w} \rangle + \sum_{i=1}^{n} \sum_{j=i+1}^{n} \langle x_i \boldsymbol{v}_i, x_j \boldsymbol{v}_j \rangle_{\boldsymbol{M}_{f_i, f_j}}, \tag{1}$$

where $w_0 \in \mathbb{R}$, $\boldsymbol{w} \in \mathbb{R}^n$, and $\boldsymbol{v}_1 \in \mathbb{R}^{k_1}$, ..., $\boldsymbol{v}_n \in \mathbb{R}^{k_n}$ are learned parameters, and the *field-interaction matrices* $\boldsymbol{M}_{f_i, f_j}$ can be either learned, predefined, or have a special structure. FMs are typically employed for supervised learning tasks on a dataset $\{(\boldsymbol{x}_i, y_i)\}_{i=1}^{N}$. Note, that this model family allows a different embedding dimension for each field, since the matrices $m\boldsymbol{M}_{f_i, f_j}$ do not have to be square. For completeness, we present the above FM variants in Appendix B, and explain why FmFMs generalize the entire family. Since FmFM generalizes the entire, we use equation 1 as the formalism to describe and prove properties which should hold for the entire family.

## 3 THE BASIS FUNCTION ENCODING APPROACH

To describe our approach we need to introduce a slight modification to the FmFM computational graph described in equation 1. Before feeding the vectors $x_i \boldsymbol{v}_i$ and scalars $x_i w_i$ into the factorization machine, we pass the vectors and, respectively, scalars belonging to each field through a linear reduction associated with that field. We consider two reductions: (a) the identity reduction, which just returns its input; and (b) a sum reduction, which sums up the $x_i \boldsymbol{v}_i$ and, respectively, the $x_i w_i$ of the corresponding field. Clearly, choosing the identity function for all fields reduces back to the regular FmFM model.

For a field $f$ that we designate as *continuous numerical*, we choose a set of functions $B_1, \ldots, B_\ell$, and encode the field as $\mathbf{enc}_f(z) = (B_1(z), \ldots, B_\ell(z))^T$ with a subsequent summing reduction. For the remaining fields, we use the identity reduction. The process for $x_i \boldsymbol{v}_i$ is depicted in Figure 2. Note, that each field can have its own basis, and in particular, the number of basis functions between fields may differ. For a formal description using matrix notation we refer the readers to Appendix A.1.

---

[1]FmFMs do not require it to be a real inner product. For it to be a true inner product, the matrix has to be square and positive definite

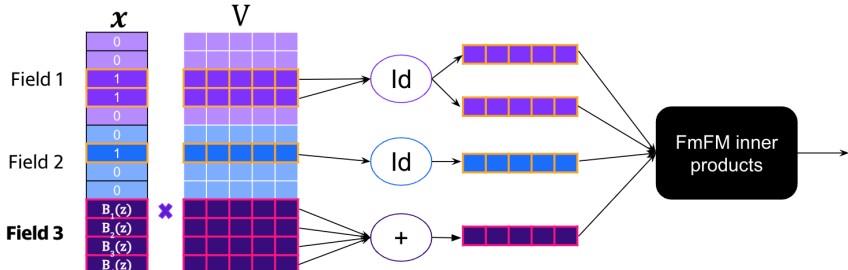

Figure 2: The computational graph with continuous numerical fields. Field 3 is a continuous numerical field whose value is $z$. The vectors $\boldsymbol{v}_1, \ldots, \boldsymbol{v}_n$ in the rows of $\boldsymbol{v}$ are multiplied by the input vector $\boldsymbol{x}$. Then, a reduction is applied to each field. Most fields have the identity (Id) reduction, whose output is identical to its input, whereas field 3 uses the sum reduction. The resulting vectors are then passed to the pairwise interaction module. An analogous process happens with the $\boldsymbol{w}$ vector.

### 3.1 SPANNING PROPERTIES

To show why our modeling choices improve the approximation power of the model, we prove two technical lemmas that establish a relationship between the basis of choice and the model's output.

**Lemma 1** (Spanning property). *Let* $\mathbf{enc}_f(z) = (B_1(z), \cdots, B_\ell(z))^T$ *be the encoding function associated with a* continuous *numerical field* $f$, *and suppose* $\phi_{\mathrm{FmFM}}$ *is computed according to equation 1. Then, assuming the remaining field values are some given constants, there exist* $\alpha_1, \ldots, \alpha_\ell, \beta \in \mathbb{R}$, *which depend on the values of the remaining fields and the model's parameters but* not *on* $z$, *such that* $\phi_{\mathrm{FmFM}}$ *as a function of* $z$ *can be written as*

$$\phi_{\mathrm{FmFM}}(z) = \sum_{i=1}^{\ell} \alpha_i B_i(z) + \beta.$$

An elementary formal proof can be found in Appendix A.2, but to convince the readers, we present an informal explanation. The vector stemming from a numerical field, after the summing reduction is a linear combination of the basis functions, while the remaining post-reduction vectors do no depend on $z$. Thus, when pairwise inner products are computed, we obtain a scalar which is a linear combination of the basis functions.

Another interesting result can be obtained by looking at $\phi_{\mathrm{FmFM}}$ as a function of *two* continuous numerical fields. It turns out that we obtain a function in the span of a tensor product of the two bases chosen for the two fields.

**Lemma 2** (Pairwise spanning property). *Let* $e, f$ *be two continuous numerical fields. Suppose that* $\mathbf{enc}_e(z_e) = (B_1(z_e), \cdots B_\ell(z_e))^T$ *and* $\mathbf{enc}_f(z_f) = (C_1(z_f), \cdots C_\kappa(z_f))^T$ *be field encoding functions, and suppose* $\phi_{\mathrm{FmFM}}$ *is computed according to equation 1. Define* $B_0(z) = C_0(z) = 1$. *Then, assuming the remaining fields are kept constant, there exist* $\alpha_{i,j}, \beta \in \mathbb{R}$ *for* $i \in \{0, \ldots, \ell\}$ *and* $j \in \{0, \ldots, \kappa\}$, *which depend on the values of the remaining fields and the model's parameters but* not *on* $z_e, z_f$, *such that* $\phi_{\mathrm{FmFM}}$ *as a function of* $z_e, z_f$ *can be written as*

$$\phi_{\mathrm{FmFM}}(z_e, z_f) = \sum_{i=0}^{\ell} \sum_{j=0}^{\kappa} \alpha_{i,j} B_i(z_e) C_j(z_f) + \beta.$$

The proof can be found in Appendix A.3. Note, that the model learns $O(\ell \cdot \kappa)$ segmentized coefficients *without* actually learning $O(\ell \cdot \kappa)$ parameters, but instead it learns only $O(\ell + \kappa)$ parameters in the form of $\ell + \kappa$ embedding vectors.

Essentially, the model learns an approximation of an optimal user behavior function for each continuous numerical field in the affine span of the chosen basis, or the tensor product basis in case of two fields. So it is natural to ask ourselves - which basis should we choose? Ideally, we should aim to choose a basis that is able to approximate functions well with a small number of basis elements, to keep training and inference efficient, and avoid over-fitting.

## 3.2 Cubic splines and the B-spline basis

Spline functions (Schoenberg, 1946) are piece-wise polynomial functions of degree $d$ with up to $d-1$ continuous derivatives defined on a set of consecutive sub-intervals of some interval $[a, b]$ defined a set of break-points $a = a_0 < a_1 < \cdots < a_{\ell-d} = b$. The case of $d = 3$ are known as *cubic splines*. Here we concentrate on splines with uniformly spaced break points, and at this stage assume that the values of our numerical field lie in a compact interval. We discuss the more generic case in the following sub-section.

It well-known that spline functions can be written as weighted sums of the celebrated B-Spline basis (de Boor, 2001, pp 87). For brevity, we will not elaborate their explicit formula in this paper, and point out that it's available in a variety of standard scientific computing packages e.g. the `scipy.interpolate.BSpline` class of the SciPy package (Virtanen et al., 2020).

It is known (de Boor, 2001, pp 149), that cubic splines can approximate an arbitrary function $g$ with $k \leq 4$ continuous derivatives up to an error bounded by $O(\|g^{(k)}\|_\infty / \ell^k)^2$, where $g^{(k)}$ is the $k^{\text{th}}$ derivative of $g$. The spanning property (Lemma 1) ensures that the model's segmentized outputs are splines spanned by the same basis, and therefore their power to approximate the optimal segmentized outputs. Therefore, assuming that the functions we aim to approximate are smooth enough and vary "slowly", in the sense that their high-order $k^{\text{th}}$ derivatives are small, the approximation error goes down at the rate of $O(\frac{1}{\ell^k})$, whereas with binning the rate is $O(\frac{1}{\ell})$.

A direct consequence is that we can obtain a theoretically good approximation which is also achievable in practice, since we can be accurate with a small number of basis functions, and this significantly decreases the chances of sparsity issues. This is in contrast to binning, where high resolution binning is required to for a good theoretical approximation accuracy, but it may not be achievable in practice.

Yet another important property of the B-Spline basis is that at any point only *four* basis functions are non-zero. Thus, regardless of the number of basis functions we use, computing the model's output remains efficient.

## 3.3 Numerical fields with arbitrary domain and distribution

Splines approximate functions on a compact interval. Thus, numerical fields with unbounded domains pose a challenge. Moreover, the support of each cubic B-Spline function is only a sub-interval of the domain defined by five consecutive knots. Thus, even if a numerical field $f$ is bounded in $[a, b]$, a highly skewed distribution may cause "starvation" of some basis functions: if $\mathbb{P}(z_f \in \text{support}(B_i))$ is extremely small, there will be little training data to effectively learn a useful representation of $B_i$.

As a remedy to both challenges, we recommend first transforming a numerical field $f$ using a function $T_f$ with two objectives: (a) map values to a compact interval, that we assume w.l.o.g. to be $[0, 1]$, and (b) spread the values such that for every $\mathbb{P}(T_f(z_f) \in \text{support}(B_i))$ is non-negligible for all $i$. In theory, if we knew the distribution of $z_f$, we could use its CDF as the transform of choice, since the transformed values would be uniformly distributed in $[0, 1]$. In practice, $T_f$ can be any function which roughly resembles the cumulative distribution function (CDF) of the field's values, as described next.

On small benchmark datasets we approximate the empirical CDF using a dense step function (see `QuantileTransform` in the Scikit-Learn package (Buitinck et al., 2013)). However, in a practical large-scale real-time recommender system, evaluating this CDF approximation is prohibitively expensive. Hence, we recommend fitting a distribution with a simple closed-form CDF, such as Normal or Student-T, to a sub-sample of the data, and using its CDF as $T_f$. Note, that our method does not eliminate the need for data analysis and feature engineering, but only simplifies it; just fit a few distributions[3], and see (visually) whose CDF resembles the empirical CDF best.

## 3.4 Integration into an existing system by simulating binning

Suppose we would like to obtain a model which employs binning of the field $f$ into a large number $N$ of intervals, e.g. $N = 1000$. As we discussed in the introduction, in most cases we cannot directly

---

[2]For a function $\phi$ defined on $S$, its infinity norm is $\|\phi\|_\infty = \max_{x \in S} |\phi(x)|$

[3]Easy to do using `scipy.stats.{some_dist}.fit()`

learn such a model because of sparsity issues. However, we can *generate* such a model from another model trained using our scheme to make initial integration easier.

The idea is best explained by referring, again, to Figure 2. For any value $z$ of the numerical field of our choice, the vector corresponding to that field after the reduction stage (field 3 in the figure) is a weighted sum of the field's vectors, where the weights are $B_i(z)$. Thus, for any field value $z$ we can compute a corresponding embedding $e(z)$. Given a set of $N$ intervals, we simply compute $N$ corresponding embedding vectors by evaluating $e$ at the mid-point of each interval. The resulting model has an embedding vector for each bin, as we desired.

The choice of the set of interval break-points may vary between fields. For example, in many practical situations a geometric progression is applicable to fields with unbounded domains. Another natural choice is utilizing the transformation $T_f$ from the previous section, that resembles the data CDF, and using its inverse computed at uniformly spaced points as the bin boundaries, i.e., $\{T_f^{-1}(\frac{j}{N})\}_{j=0}^{N}$.

## 4 EVALUATION

We divide this section into three parts. First, we use a synthetically generated data-set to show that our theory holds - the model learns segmentized output functions that resemble the ground truth. Then, we compare the accuracy obtained with binning versus splines on several data-sets. The code to reproduce these experiments is available in the supplemental material. Finally, we report the results of a web-scale A/B test conducted on a major online advertising platform.

### 4.1 LEARNING ARTIFICIALLY CHOSEN FUNCTIONS

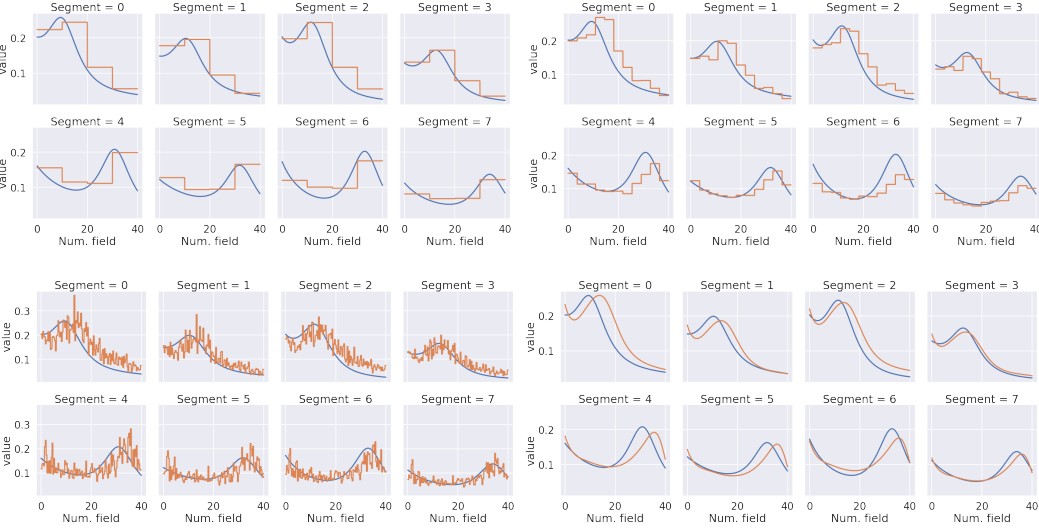

Figure 3: Results of segmentized approximations of four FFM models trained on synthetic data, with 5, 12, and 120 bins, and splines with 6 break-points, down-left order. 5 bins have poor accuracy (test loss = 0.3474), 12 bins have a better accuracy (test loss = 0.3442), with 120 bins the accuracy worsens due to sparsity (test loss = 0.3478), and with splines we achieve best accuracy (0.3432).

We used a synthetic toy click-through rate prediction data-set with four fields, and zero-one labels (click / non-click). Naturally, the cross-entropy loss is used to train models on such tasks. We have three categorical fields each having two values each, and one numerical field in the range $[0, 40]$. For each of the eight segment configurations defined by the categorical fields, we defined functions $p_0, \ldots, p_7$ (see Figure 3) describing the CTR as a function of the numerical field. Then, we generated a data-set of 25,000 rows, such that for each row $i$ we chose a segment configuration $s_i \in \{0, ..., 7\}$ of the categorical fields uniformly at random, the value of the numerical field $z_i \sim \text{Beta-Binomial}(40, 0.9, 1.2)$, and a label $y_i \sim \text{Bernoulli}(p_{s_i}(z_i))$.

We trained an FFM (Juan et al., 2016) provided by Yahoo-Inc (2023) using the binary cross-entropy loss on the above data, both with binning of several resolutions and with splines defined on 6 sub-intervals. The numerical field was naïvely transformed to $[0, 1]$ by simple normalization. We plotted the learned curve for every configuration in Figure 3. Indeed, low resolution binning approximates poorly, a higher resolution approximates better, and a too-high resolution cannot be learned because of sparsity. However, Splines defined on only six sub-intervals approximate the synthetic functions $\{p_i\}_{i=0}^7$ quite well.

Next, we compared the test cross-entropy loss on 75,000 samples generated in the same manner with for several numbers of intervals used for binning and cubic Splines. For each number of intervals we performed 15 experiments to neutralize the effect of random model initialization. As is apparent in Figure 4, Splines consistently outperform in this theoretical setting. The test loss obtained by both strategies diminishes if the number of intervals becomes too large, but the effect is much more significant in the binning solution.

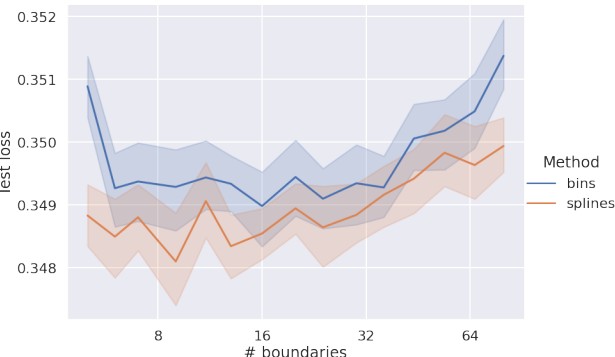

Figure 4: Comparison of the test cross-entropy loss obtained with Splines and bins. Both methods suffer from sparsity issues as the number of intervals grows, but Splines are able to utilize their approximation power with a small number of intervals, before sparsity takes effect. The bands are bootstrap CIs - for each number of boundaries and numerical feature type we ran 15 experiments to neutralize the effect of random initialization.

## 4.2 PUBLIC DATA-SETS

We test our approach versus binning on several tabular data-sets: the california housing (Pace & Barry, 1997) , adult income (Kohavi, 1996), Higgs (Baldi et al., 2014) (we use the 98K version from OpenML (Vanschoren et al., 2014)), and song year prediction (Bertin-Mahieux et al., 2011). For the first two data-sets we used an FFM, whereas for the last two we used an FM, both provided by Yahoo (Yahoo-Inc, 2023), since an FFM is more expensive when there are many columns. We used Optuna (Akiba et al., 2019) for hyper-parameter tuning, e.g. step-size, batch-size, number of intervals, and embedding dimension, separately for each strategy. For binning, we also tuned the choice of uniform or quantile bins. In addition, 20% of the data was held out for validation, and regression targets were standardized. Finally, for the adult income data-set, 0 has a special meaning for two columns, and was treated as a categorical value.

We ran 20 experiments with the tuned configurations to neutralize the effect of random initialization, and report the mean and standard deviation of the metrics on the held-out set in Table 1, where it is apparent that our approach outperforms binning on these datasets. These datasets were chosen since they contain several numerical fields, and are small enough to run a many experiments to neutralize the effect of hyper-parameter choice and random initialization at a reasonable compute cost, or time. They were also used in other works on tabular data, such as Gorishniy et al. (2021; 2022).

## 4.3 A/B TEST RESULTS ON AN ONLINE ADVERTISING SYSTEM

Here we report an online performance improvement measured using an A/B test, serving real traffic of a major online advertising platform. The platform applies a proprietary FM family *click-through-rate* (CTR) prediction model that is closely related to FwFM (Sun et al., 2021). The model, which provides

|  | Cal. housing | Adult income | Higgs | Song year |
|---|---|---|---|---|
| # rows | 20640 | 48842 | 98050 | 515345 |
| # cat. fields | 0 | 8 | 0 | 0 |
| # num. fields | 8 | 6 | 28 | 88 |
| Metric type | RMSE | Cross-Entropy | Cross-Entropy | RMSE |
| Binning metric (std%) | 0.4730 (0.36%) | 0.2990 (0.47%) | 0.5637 (0.22%) | 0.9186 (0.4%) |
| Splines metric (std%) | **0.4294** (0.57%) | **0.2861** (0.28%) | **0.5448** (0.12%) | **0.8803** (0.2%) |
| Splines vs. binning | $-9.2\%$ | $-4.3\%$ | $-3.36\%$ | $-4.16\%$ |

Table 1: Comparison of binning vs. splines. Standard deviations are reported as % of the mean.

CTR predictions for merchant catalogue-based ads, has a *recency* field that measures the time (in hours) passed since the user viewed a product at the advertiser's site. We compared an implementation using our approach of continuous feature training and high-resolution binning during serving time described in Section 3.4 with a fine grained geometric progression of 200 bin break points, versus the "conventional" binned training and serving approach used in the production model at that time. The new model is only one of the rankers[4] that participates in our ad auction. Therefore, a mis-prediction means the ad either unjustifiably wins or loses the auction, both leading to revenue losses.

We conducted an A/B test against the production model at that time, when our new model was serving $40\%$ of the traffic for over six days. The new model dramatically reduced the CTR prediction error, measured as $(\frac{\text{Average predicted CTR}}{\text{Measured CTR}} - 1)$ on an hourly basis, from an average of $21\%$ in the baseline model, to an average of $8\%$ in the new model. The significant increase in accuracy has resulted in this model being adopted as the new production model.

## 5 DISCUSSION

We presented an easy to implement approach for improving the accuracy of the factorization machine family whose input includes numerical features. Our scheme avoids increasing the number of model parameters and introducing over-fitting, by relying on the approximation power of cubic splines. This is explained by the spanning property together with the spline approximation theorems. Moreover, the discretization strategy described in Section 3.4 allows our idea to be integrated into an existing recommendation system without introducing major changes to the production code that utilizes the model to rank items.

It is easy to verify that our idea can be extended to factorization machine models of higher order (Blondel et al., 2016). In particular, the spanning property in Lemma 1 still holds, and the pairwise spanning property in Lemma 2 becomes $q$-wise spanning property from machines of order $q$. However, to keep the paper focused and readable, we keep the analysis out of the scope of this paper.

With many advantages, our approach is not without limitations. We do not eliminate the need for data research and feature engineering, which is often required when working with tabular data, since the data still needs to be analyzed to fit a function that roughly resembles the empirical CDF. Feature engineering becomes somewhat easier, but some work still has to be done.

Finally, we would like to note two drawbacks. First, our approach slightly reduces interpretability, since we cannot associate a feature with a corresponding learned latent vector. Second, our approach may not be applicable to all kinds of numerical fields. For example, consider a product recommendation system with a product price field. Usually higher prices mean a different category of products, leading to a possibly different trend of user preferences. In that case, the optimal segmentized output as a function of the product's price is probably far from having small (higher order) derivatives, and thus cubic splines may perform poorly, and possibly even worse than simple binning.

---

[4]Usually each auction is conducted among several models (or "rankers"), that rank their ad inventories and compete over the incoming impression.

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

# A  PROOF OF THE SPANNING PROPERTIES

First, we write the FmFM model using matrix notation. Then, we use the matrix notation to prove our Lemmas.

## A.1  FORMALIZATION USING LINEAR ALGEBRA

To formalize our approach, we denote by $\boldsymbol{v}$ the matrix whose *rows* are the vectors $\boldsymbol{v}_i$, and decompose the FmFM formula equation 1 into three sub-formulas:

$$\boldsymbol{y} = \operatorname{diag}(\boldsymbol{x})\boldsymbol{w},$$
$$\boldsymbol{P} = \operatorname{diag}(\boldsymbol{x})\boldsymbol{v},$$
$$\phi_{\mathrm{FmFM}} = w_0 + \langle \mathbf{1}, \mathbf{y} \rangle + \sum_{i=1}^{n} \sum_{j=i+1}^{n} \langle \boldsymbol{P}_i, \boldsymbol{P}_j \rangle_{M_{f_i, f_j}},$$

where the $\operatorname{diag}(\cdot)$ operator creates a diagonal matrix with the argument on the diagonal, $\mathbf{1}$ is a vector whose components are all 1, and $\boldsymbol{P}_i$ is the $i$th row of $\boldsymbol{P}$. Next, we associate each field $f$ with a *field reduction* matrix $\mathbf{R}_f$, concatenate them into one big block-diagonal reduction matrix. Note, that neither the block matrices $R_f$, nor the matrix $R$ have to be square.

$$\mathbf{R} = \begin{bmatrix} \mathbf{R}_1 & \mathbf{0} & \dots & \mathbf{0} \\ \mathbf{0} & \mathbf{R}_2 & \dots & \mathbf{0} \\ \vdots & \vdots & \ddots & \vdots \\ \mathbf{0} & \mathbf{0} & \dots & \mathbf{R}_m \end{bmatrix},$$

and assuming $\mathbf{R}$ has $\hat{n}$ rows, we modify the FmFM formula as:

$$\hat{\boldsymbol{y}} = \mathbf{R} \operatorname{diag}(\boldsymbol{x})\boldsymbol{w},$$
$$\hat{\boldsymbol{P}} = \mathbf{R} \operatorname{diag}(\boldsymbol{x})\boldsymbol{v},$$
$$\phi_{\mathrm{FmFM}} = w_0 + \langle \mathbf{1}, \hat{\boldsymbol{y}} \rangle + \sum_{i=1}^{\hat{n}} \sum_{j=i+1}^{\hat{n}} \langle \hat{\boldsymbol{P}}_i, \hat{\boldsymbol{P}}_j \rangle_{M_{f_i, f_j}}. \tag{2}$$

Setting $\mathbf{R}_f = \mathbf{I}$ for a field $f$ amounts to the identity reduction, whereas setting $\mathbf{R}_f = \mathbf{1}^T$, will cause the scalars $x_i w_i$ and the vectors $x_i \boldsymbol{v}_i$ to be summed up, resulting in the summing reduction. This matrix notation is useful for the study of the theoretical properties of our proposal, but in practice we will apply the field reductions manually, as efficiently as possible without matrix multiplication.

## A.2  PROOF OF THE SPANNING PROPERTY (LEMMA 1)

*Proof.* Recall that we need to rewrite equation 2 as a function of $z$. For some vector $\mathbf{q}$, denote by $\mathbf{q}_{a:b}$ the sub-vector $(q_a, \dots, q_b)$. Assume w.l.o.g. that $f = 1$, and that field 1 has the value $z$. By construction in equation 2, we have $\hat{y}_1 = \sum_{i=1}^{\ell} w_i B_i(z)$, while the remaining components $\hat{\boldsymbol{y}}_{2:\hat{n}}$ do not depend on $z$. Thus, the we have

$$w_0 + \langle \mathbf{1}, \hat{\boldsymbol{y}} \rangle = \sum_{i=1}^{\ell} w_i B_i(z) + \underbrace{w_0 + \langle \mathbf{1}, \hat{\boldsymbol{y}}_{2:\hat{n}} \rangle}_{\beta_1}. \tag{3}$$

Moreover, by equation 2 we have that $\hat{P}_1 = \sum_{i=1}^{\ell} v_i B_i(z)$, whereas the remaining rows $\hat{P}_2, \ldots, \hat{P}_{\hat{n}}$ do not depend on $z$. Thus,

$$
\begin{aligned}
\sum_{i=1}^{\hat{n}} \sum_{j=i+1}^{\hat{n}} \langle \hat{P}_i, \hat{P}_j \rangle_{M_{f_i,f_j}} &= \sum_{j=2}^{\hat{n}} \langle \hat{P}_1, \hat{P}_j \rangle_{M_{1,f_j}} + \underbrace{\sum_{i=2}^{\hat{n}} \sum_{j=i+1}^{\hat{n}} \langle \hat{P}_i, \hat{P}_j \rangle_{M_{f_i,f_j}}}_{\beta_2} \\
&= \sum_{j=2}^{\hat{n}} \langle \sum_{i=1}^{\ell} v_i B_i(z), \hat{P}_j \rangle_{M_{1,f_j}} + \beta_2 \\
&= \sum_{i=1}^{\ell} \underbrace{\left( \sum_{j=2}^{\hat{n}} \langle v_i, \hat{P}_j \rangle_{M_{1,f_j}} \right)}_{\tilde{\alpha}_i} B_i(z) + \beta_2.
\end{aligned}
\tag{4}
$$

Combining equation 3 and equation 4, we obtain

$$
\phi_{\text{FmFM}}(z) = \sum_{i=1}^{\ell} (w_i + \tilde{\alpha}_i) B_i(z) + (\beta_1 + \beta_2),
$$

which is of the desired form. $\qquad\square$

### A.3 PROOF OF THE PAIRWISE SPANNING PROPERTY (LEMMA 2)

*Proof.* Recall that we need to rewrite equation 2 as a function of $z_e, z_f$, which are the values of the fields $e$ and $f$. Assume w.l.o.g. that $e = 1, f = 2$. By construction in equation 2, we have $\hat{y}_1 = \sum_{i=1}^{\ell} w_i B_i(z_1)$, and $\hat{y}_2 = \sum_{i=1}^{\kappa} w_i C_i(z_2)$, while the remaining components $\hat{y}_{3:\hat{n}}$ do not depend on $z$. Thus, the we have

$$
\begin{aligned}
w_0 + \langle \mathbf{1}, \hat{y} \rangle &= \sum_{i=1}^{\ell} w_i B_i(z_1) + \sum_{i=1}^{\kappa} w_{i+\ell} C_i(z_2) + \underbrace{w_0 + \langle \mathbf{1}, \hat{y}_{3:\hat{n}} \rangle}_{\beta_1} \\
&= \sum_{i=0}^{\ell} \sum_{j=0}^{\kappa} \hat{\alpha}_{i,j} B_i(z_1) C_j(z_2) + \beta_1,
\end{aligned}
\tag{5}
$$

where $\hat{\alpha}_{i,0} = w_i, \hat{\alpha}_{0,j} = w_{i+\ell}$ for all $i, j \geq 1$, for all other values of $i, j$ we set $\hat{\alpha}_{i,j} = 0$.

Moreover, by equation 2 we have that $\hat{P}_1 = \sum_{i=1}^{\ell} v_i B_i(z_1)$ and $\hat{P}_2 = \sum_{i=1}^{\kappa} v_i C_{i+\ell}(z_2)$, whereas the remaining rows $\hat{P}_3, \ldots, \hat{P}_{\hat{n}}$ do not depend on $z_1, z_2$. First, let us rewrite the interaction between $z_1, z_2$ specifically:

$$
\langle p_1, p_2 \rangle_{M_{1,2}} = \langle \sum_{i=1}^{\ell} v_i B_i(z), \sum_{i=1}^{\kappa} v_{i+\ell} C_i(z) \rangle_{M_{1,2}} = \sum_{i=1}^{\ell} \sum_{j=1}^{\kappa} \underbrace{\langle v_i, v_{j+\ell} \rangle_{M_{1,2}}}_{\gamma_{i,j}} B_i(z_1) C_j(z_2)
\tag{6}
$$

By following similar logic to 4, one can obtain a similar expression when looking at the partial sums that include the interaction between $f$ (resp. $e$) and all other fields except $e$ (resp. $f$). Observe that the interaction between all other values does not depend on $z_1, z_2$. Given all of this, we show how to

rewrite the interaction as a function of $z_1, z_2$:

$$
\sum_{i=1}^{\hat{n}} \sum_{j=i+1}^{\hat{n}} \langle \hat{\boldsymbol{P}}_i, \hat{\boldsymbol{P}}_j \rangle_{M_{f_i, f_j}}
$$

$$
= \langle \hat{\boldsymbol{P}}_1, \hat{\boldsymbol{P}}_2 \rangle_{M_{1,2}} + \sum_{j=3}^{\hat{n}} \langle \hat{\boldsymbol{P}}_1, \hat{\boldsymbol{P}}_j \rangle_{M_{1,f_j}} + \sum_{j=3}^{\hat{n}} \langle \hat{\boldsymbol{P}}_2, \hat{\boldsymbol{P}}_j \rangle_{M_{2,f_i}} + \underbrace{\sum_{i=3}^{\hat{n}} \sum_{j=i+1}^{\hat{n}} \langle \hat{\boldsymbol{P}}_i, \hat{\boldsymbol{P}}_j \rangle_{M_{f_i, f_j}}}_{\beta_2} \quad (7)
$$

$$
= \sum_{i=1}^{\ell} \sum_{j=1}^{\kappa} \gamma_{i,j} B_i(z_1) C_j(z_2) + \sum_{i=1}^{\ell} \tilde{\alpha}_i B_i(z_1) + \sum_{i=1}^{\kappa} \bar{\alpha}_i C_i(z_2) + \beta_2
$$

where $\tilde{\alpha}_i, \bar{\alpha}_i$ are obtained similarly in equation 4's final step.

$\square$

## B  THE FACTORIZATION MACHINE FAMILY

Factorization machines are formally described as models whose input is a feature vector $\boldsymbol{x}$ representing rows in tabular data-sets, as described in section 2. In the context of recommendation systems, the data-set contains past interactions between users and items, whose columns, often named *fields*, and whose values are *features*. The columns describe the context, such as the user's gender and age, the time of visit, or the article the user is currently reading, whereas others describe the item, such as product category, or item popularity.

The initial Factorization Machines (FMs), as proposed in Rendle (2010), compute

$$
\Phi_{\text{FM}}(\boldsymbol{x}) = w_0 + \langle \boldsymbol{x}, \boldsymbol{w} \rangle + \sum_{i=1}^{n} \sum_{j=i+1}^{n} \langle x_i \boldsymbol{v}_i, x_j \boldsymbol{v}_j \rangle.
$$

The learned parameters are $w_0 \in \mathbb{R}$, $\boldsymbol{w} \in \mathbb{R}^n$, and $\boldsymbol{v}_1, \ldots, \boldsymbol{v}_n \in \mathbb{R}^k$, where $k$ is a hyper-parameter. The model can be thought of as a way to represent the quadratic interaction model

$$
\Phi_{\text{quad}}(\boldsymbol{x}) = w_0 + \langle \boldsymbol{x}, \boldsymbol{w} \rangle + \sum_{i=1}^{n} \sum_{j=i+1}^{n} A_{i,j} x_i x_j,
$$

where the coefficient matrix $\boldsymbol{A}$ is represented in factorized form. The vectors $\boldsymbol{v}_1, \ldots, \boldsymbol{v}_n$ are the feature embedding vectors. Classical matrix factorization is recovered when we have only user id and item id fields, whose values are one-hot encoded. We note that this is a special case of the FmFM model in equation 1, with $\boldsymbol{M}_{f_i, f_j} = \boldsymbol{I}$.

The $\Phi_{\text{FM}}$ model does not represent the varying behavior of a feature belonging to some field when interacting with features from different fields. For example, genders may interact with ages differently than they interact with product categories. Initially, to explicitly encode this information into a model, the Field-aware Factorization Machine (FFM) was proposed in Juan et al. (2016). The each embedding vector $\boldsymbol{v}_i$ is modeled as a concatenation of field-specific embedding vectors for each of the $m$ fields:

$$
\boldsymbol{v}_i = \begin{bmatrix} \boldsymbol{v}_{i,1} \\ \vdots \\ \boldsymbol{v}_{i,m} \end{bmatrix},
$$

where $\boldsymbol{v}_{i,f}$ is the embedding vector of feature $i$ when interacting with another feature from a field $f$. The model computes

$$
\Phi_{\text{FFM}}(\boldsymbol{x}) = w_0 + \langle \boldsymbol{x}, \boldsymbol{w} \rangle + \sum_{i=1}^{n} \sum_{j=i+1}^{n} \langle x_i \boldsymbol{v}_{i,f_j}, x_j \boldsymbol{v}_{j,f_i} \rangle
$$

For any field $f$, let $\boldsymbol{P}_f$ be the matrix that extracts $\boldsymbol{v}_{i,f}$ from $\boldsymbol{v}_i$, namely, $\boldsymbol{P}_f \boldsymbol{v}_i = \boldsymbol{v}_{i,f}$. Then FFMs are also a special case of the FmFM model in equation 1 with $\boldsymbol{M}_{e,f} = \boldsymbol{P}_f^T \boldsymbol{P}_e$.

As pointed out by Pan et al. (2018); Juan et al. (2016), the FFM models are prone to over-fitting, since it learns a feature embedding vector for each feature x field pair. As a remedy, Pan et al. (2018) proposed the Field-Weighted Factorization Machine (FwFM) that models the varying behavior of field interaction using learned scalar field interaction intensities $r_{e,f}$ for each pair of fields $e, f$. The FwFM computes

$$\Phi_{\text{FwFM}}(\boldsymbol{x}) = w_0 + \langle \boldsymbol{x}, \boldsymbol{w} \rangle + \sum_{i=1}^{n} \sum_{j=i+1}^{n} r_{i,j} \langle x_i \boldsymbol{v}_i, x_j \boldsymbol{v}_j \rangle.$$

Letting $\boldsymbol{M}_{e,f} = r_{i,j} \boldsymbol{I}$, we recover the FmFM model in equation 1.

