# OpenReview forum: "Basis Function Encoding of Numerical Features in Factorization Machines for Improved Accuracy"
_ICLR.cc/2024/Conference — Submitted to ICLR 2024_

### Official Review · Reviewer_cQDt · 2023-11-01

**Soundness:** 3 good
**Presentation:** 4 excellent
**Contribution:** 3 good
**Rating:** 6
**Confidence:** 4

**Summary:**

The authors present a method for incorporating b-spline interpolation into a factorization machine (FM) framework for online content recommendation systems. Factorization machines have attractive properties in high-throughput production recommendation systems where inference speed is needed. Existing factorization machine frameworks use binning, which amounts to an overly crude, non-smooth interpolation approach.  Basis function interpolation is proposed as an alternative which can fit into the overall FM approach, with b-splines being the basis used in the paper. The b-spline FMs can be mapped onto a binning approach with many bins if that mapping is convenient for existing production software systems.  The approach is illustrated on toy data, on a suite of modest-size real world tabular data and (most importantly) in an A/B test of a major online advertising platform in which a dramatic improvement was achieved, with click-through-rate prediction error dropping from 21% for the previous system to 8% with the b-spline FM approach.

**Strengths:**

The reported AB test improvement for a major advertising platform seems to me to be very strong evidence. The only counterargument I can think of is that maybe the preexisting system was not well executed and was not using a state-of-the-art model, but this seems unlikely in the setting of a major advertising platform.

The work is original to the best of my knowledge. I was previously familiar with the basics of factorization machines, but I'm not an expert on them, so I'm not 100% sure the work is original, but as far as I know, it is.

The paper is extremely well written. I caught a couple of typos which I will list in the weaknesses section, but overall, the paper was very easy to read and understand.

As I'll say in the weaknesses, I would like a little more info on why factorization machines are needed rather than matrix factorization and how widespread factorization machines really are in recommendation systems. That's my one hesitation regarding significance. If I can be reassured in that area, I'd feel even better about the significance and would raise my score.

**Weaknesses:**

My one concern is that the paper seems to take an attitude of "everyone knows factorization machines are widely used in recommendation systems" and I'm not sure that's as widely known as the authors think it is. Related to that, I would like understand why matrix factorization is not appropriate for the recommendation problem the authors are concerned with at the advertising platform. The paper doesn't even mention matrix factorization. It surely deserves at least a brief mention so that readers understand why it's not the best solution here.

I probably represent something like the median reader in terms of pre-existing familiarity with the topic. I do not work actively in recommendation systems, but I once published a recommendation systems paper (one based on matrix factorization) about a decade ago.  A Google Scholar search for  ' "matrix factorization" recommendation systems' yields about 10X more results than ' "factorization machines" recommendation systems and the Yehuda Koren paper on matrix factorization is much more widely cited than the original Rendle factorization machine paper. So I was a bit surprised that the paper basically seems to take it for granted that a reader would understand that factorization machines are commonly used in recommendation systems.

Relatedly, I would like to hear more detailed examples of what the fields typically are.  On the bottom of page 1, the authors say the fields are "past interactions between users and items" but the language elsewhere in the paper makes it sound like it's often just descriptors of the item and/or user rather than data on the interaction between them.

Granted, factorization machines and matrix factorization are a bit similar in spirit in that they both represent interaction parameters with low-rank matrices, i.e, dot products between vectors of free parameters, but they're fundamentally different approaches.FM is essentially polynomial regression with the coefficients of the regression being constrained to live in a low-rank subspace so as to overcome the curse of dimensionality, with the regression predictors being encodings of fields, as the authors describe. On the other hand, in (the simpled form of ) matrix factorization, each user and each item is represented as a vector of parameters (to be fit in training) and the interaction between user and item is represented as the dot product between the 2 vectors. So why is this approach not appropriate for the problem the authors tackle?


Typos:
Page 4 weather the bins -> whether the bins
Page 5 Lemma 1: Equation equation 1 -> equation 1
Page 5 do no depend -> do not depend
Page 5 suppose that … be -> suppose that … are
Page 8 closely related FwFM - > closely related to FwFM


**** Update after rebuttal *****

I appreciate the explanation regarding the motivation for using FM instead of MF. Nonetheless, I will keep my score the same. I agree with reviewer zL9t that the absence of Criteo benchmarking is a concern.

**Questions:**

Can you describe what the fields are in a little more detail ? Are they really data on interactions between users and items, or they just descriptors of users and/or items?

Why is matrix factorization not appropriate for the online advertising application?

I am open to raising my score if my questions are answered adequately.

---

> ### Author Response · Authors · 2023-11-17
>
> We kindly ask the reviewer to read the general comment at the top, before reading the answers.
>
> Regarding the questions:
> 1. Fields are columns in the tabular data-set. For example 'age' or 'ad category' are fields, whereas the values in the columns, such as '20 years old' or 'finance ad' are features. We also clarified it in the second paragraph in the introduction in the latest upload.
> 2. MF may be appropriate when we want to factorize the logit matrix for each user and ad. But we usually want to utilize user and item features, rather than their identity, to overcome sparsity and cold start challenges, as we explain in the general comment at the top of the page. We also clarified the difference between FM and MF in the paper, as pointed out in the comment.

---

> > ### Comment · Reviewer_cQDt · 2023-11-20
> > **Thank you for the responses**
> >
> > I appreciate the explanation regarding the motivation for using FM instead of MF. Nonetheless, I will keep my score the same. I agree with reviewer zL9t that the absence of Criteo benchmarking is a concern.

---

### Official Review · Reviewer_TbNK · 2023-11-08

**Soundness:** 2 fair
**Presentation:** 2 fair
**Contribution:** 2 fair
**Rating:** 5
**Confidence:** 2

**Summary:**

This paper tackles the problem of encoding numerical features in factorization machines.
The authors proposed to use B-splines to encode numerical features in factorization machines. Theoretical analysis shows that this feature map learns a segmentized function, where the coefficients depend on remaining fields, as opposed to traditional binning feature map. The authors conducted simulation studies to show  the proposed model can learn smooth functions. Then, several experiments on public datasets are conducted. Finally, the authors reported results from an A/B test in the real-world system.

**Strengths:**

1. The motivation seems reasonable, since  the binning transforms loss information and do not work for sparse data. Simulation studies also prove the issue.
2. Both simulation studies and real-world experiments are conducted, including an online A/B test.

**Weaknesses:**

1. The proposed B-spline feature map seems not new and similar approaches are studied in other fields. For example, the similar random Fourier features are studied theoretically [1] and applied broadly in fields like NeRF, positional encoding in Transformers and diffusion models. I am wondering what are the differences and advantages of the proposed feature map.
- [1]. Tancik, et al. (2020). Fourier features let networks learn high frequency functions in low dimensional domains. NeurIPS.

2. The authors only compared the binning feature map, which seems to be a very classical baseline. Is there any model aiming to solve the similar issue of binning feature  map? It would be more convincing to compare with some other feature maps and other FMs, e.g., mentioned in the related work.

**Questions:**

1. According to Section 3.3, there are actually two feature maps, the proposed spline $f$ and another continuous function $T_f$. Is there any ablation study to show the effect of these two transforms? For example, remove the B-spline and only preserve the continuous function $T_f$.

2. The authors mentioned the computational efficiency of the proposed approach. Is there a comparison about the computing time?

---

> ### Author Response · Authors · 2023-11-17
>
> We kindly ask the reviewer to read the general comments at the top. they answers question (3) and some of the concerns raised in the weaknesses.
>
> Regarding computational efficiency, we indeed have not conducted such studies. But it heavily depends on having a very efficient implementation that integrates well with the rest of the code with little overhead.
>
> If it's interesting for you, in the product where we conducted the A/B tests reported in the paper there was no visible effect on ranking latency, and ~2% increase in the time it takes to execute the training pipeline. But this is not very informative, since evaluating the cubic B-Spline basis and adding up the four vectors in the corresponding field is just one of the many things the pipeline does.

---

> > ### Comment · Reviewer_TbNK · 2023-11-21
> >
> > Thanks for the authors’ reply. I understand that using the spline method is for computational efficiency. The online A/B test proves the proposed approach works well in practice. However, as a research paper, I think the motivation and contribution are still limited for ICLR. Moreover, as the authors said, the A/B test results may be affected by many things. So the empirical results may not fully support their contributions. Therefore, I would maintain my score.

---

### Official Review · Reviewer_zL9t · 2023-11-08

**Soundness:** 3 good
**Presentation:** 2 fair
**Contribution:** 3 good
**Rating:** 6
**Confidence:** 3

**Summary:**

The authors introduce a method for encoding numerical features in factorization machines (FMs) to increase the model's expressive power without an excessive increase in the number of parameters which need to be learned (which leads to sparsity issues with finite data). Rather than a one-hot encoding of the numerical feature according to a binning scheme, they instead encode it using basis functions: $z \mapsto (B_1(z),\ldots,B_\ell(z))$. They theoretically quantify the expressive power of the resulting model. Finally, they conduct extensive experiments on synthetic and real data to empirically confirm their method's improved performance.

**Strengths:**

The authors tackle a relevant problem in the field of tabular ML. In the age of foundation models, there is still a practical need for models which can be applied to situations where fast training and inference are crucial.

The theoretical component is sound. I checked the proofs in the appendix and did not find any errors. The theoretical results also give a clear picture of the advantages of the proposed method. For instance, a natural question in this setting is, if we desire to have a function class which is more expressive for continuous features, can we simply *not* discretize the numerical feature to avoid ending up with a piecewise continuous function? What is the benefit of encoding it with basis functions? Their lemmas answer this question: this is equivalent to "encoding" the feature using the single basis function $B_1(z) = z$. Their results show that the resulting function must be linear in the numerical feature when all other features are held constant. I also found the discussion of the results, and some "informal" theoretical motivation (e.g., that the approximation error should be $O(1/\ell^k)$ for their method vs. $O(1/\ell)$ for naive binning, where $\ell$ is the number of bins/break points and $k\leq 3$ is the number of continuous derivatives of the function to be approximated) to be helpful.

The empirical results are also solid. They conduct experiments on synthetic data, four publicly available real datasets, and a proprietary real-world click-through rate setting, all of which show gains for their method over the binning approach.

There is also a candid discussion of the drawbacks of the proposed method, including a decrease in interpretability as compared to binning, and situations in which the user may expect the proposed method to actually have *worse* performance as compared to binning. This discussion improves the usability of the method.

Finally, I am not an expert on FMs, but at least based on the cited related work, the contribution is novel.

**Weaknesses:**

**Empirical Results:** In the synthetic experiments, the test loss for the proposed method is not statistically significantly lower than that of the binning method. Given that it seems that the function used to generated the data matches the cubic splines very closely (see also the Questions section below), this is somewhat surprising. If it is just a case of a small number of trials (15), the authors should run more experiments to show that there is a significant separation.

The empirical results on real-world advertising data are certainly impressive. However, since both the base algorithm ("a proprietary FM family click-through-rate (CTR) prediction model that is closely related to FwFM", pg. 8) and the data are proprietary, these results cannot be verified. Any additional information that can be provided on these results, such as the scale of the dataset or measures of uncertainty on the difference in accuracy, would be helpful.

**Presentation:** There are some typos and other presentation issues that detract from the paper. For example:
- The term "segmentized function" is used throughout the paper. I think this means a piecewise function, but I didn't see this defined explicitly anywhere.
- The term "field" is used to refer to a component of the base tabular dataset. Since there is a significant theoretical component to the paper, this may be confused with a field in the mathematical sense, so it should be defined explicitly.
- In Section 3.4, there is a "plain English" explanation of how the method can be integrated into an existing binning-based pipeline. A mathematically precise algorithm description would be helpful to understand this procedure (see the Questions section below).
- "easier" is repeated twice in the sentence beginning "Moreover, to make integration..." (second to last paragraph, pg. 2)
- Double period in point (a) of the summary paragraph (last paragraph, pg. 2)
- "Equation" is often repeated when referencing an equation number, e.g. first sentence of Section 3 (pg. 4), statement of Lemma 1 (pg. 5)
- There is a minor notational inconsistency in Fig. 2: presumably, the "rows of $v$" in the caption refers to the matrix $V$ in the figure.
- The "matrix" $\mathbf{P}$ defined in the appendix is not really a proper matrix, since the rows do not all have the same dimension. Defining the individual vectors $\mathbf{P}_i$ would be more mathematically conventional than stacking them into a jagged matrix, but this is minor.

This is not a comprehensive list. The paper should be carefully edited before publication.

**Questions:**

1. Can the authors clarify the explanation in Section 3.4? My understanding is as follows: The binning-style model requires an embedding vector for each bin. Instead of trying to learn these directly from data, which will have sparsity issues, we first learn using the basis function approach. We then compute the embedding vector output by the basis function model at the midpoint of each bin, and use these as the "learned" embedding vectors for the binning model. Is this correct?

2. What are the functions $p_i$ in the synthetic experiment? They seem to be a very similar shape to the cubic splines; are these functions themselves cubic splines? If so, is the reason for the error of the proposed method just due to a finite training set?

3. A more general question about FMs: Based on equation (1), it seems like an FM is a feature embedding + a second order polynomial regression, where there are no "diagonal" (i.e., those of the form $x_i^2$) second order terms. Perhaps there are some other restrictions on the coefficients that one can get for the resulting quadratic, based on the form of (1). The intro did a good job explaining why these methods are preferred over larger models like neural networks when fast inference is a necessity. My question is, is there a specific benefit or motivation for this particular form of model which makes it preferable to a general quadratic function? Either a discussion of this or a pointer to a specific reference would make the paper more accessible to readers who are unfamiliar with FMs.

---

> ### Author Response · Authors · 2023-11-16
> **Answers to questions**
>
> We kindly ask you to read the general comment above first. Below we address your questions:
> 1. Your understanding is correct. That's exactly what we are proposing
> 2. The functions $p_i$ are of the form $\frac{a}{(a + b + x)^{1.5}} + \frac{a}{(a + b)^{1.5}} \frac{5}{\cosh(x - \beta)}$, where $a, b, \beta$ are parameters. We wanted a function with a "bump" somewhere that we can move and change its height using the parameters. Since we put the code to reproduce all experimental results, interested readers may download the supplemental material and see exactly what these functions are. If you believe it’s essential in the paper itself, we’ll add it to the figure’s description (you can find it in the ctr_curve_learning notebook, cells 5-7)
> 3. See our general comment, part (3). It’s “almost” a general quadratic function, but in factorized form. The reason the diagonal is missing is simple - the effect of single features is already captured by the linear part, so the quadratic part’s purpose is modeling the effect of pairs of features. We hope that what we added in the appendix which we referred to from Section 2.1 indeed makes the paper more accessible. Please see the 2010 paper 'Factorization Machines' by Stephen Rendle for more details.

---

> > ### Comment · Reviewer_zL9t · 2023-11-17
> >
> > Thanks to the authors for their response. I've also looked at the other reviewers' comments and author responses.
> >
> > **Strengths after the response:**
> > 1. A common complaint was that the authors did not compare against neural network based approached. I believe the authors have adequately explained why this is out of scope for the paper (training/inference latency issues).
> > 2. I do think it would be helpful to include the description of the $p_i$ in the figure, since it shows readers that the target isn't chosen to make it artificially easy for the proposed method but harder for the baselines.
> > 3. The additional background on FMs that the authors have added in the paper/appendix provides some helpful clarity for non-experts. I appreciate this addition and believe it has strengthened the paper.
> >
> > **Weaknesses after the response:**
> > 1. There are still no additional details on the real advertising data experiment. I understand that researchers working in industry are constrained in terms of what details they can disclose. However, as ICLR is an academic research conference, and the strength of this empirical result seems to be the main selling point of the paper, this still raises concerns for me since the result cannot be verified (as I mentioned in my original review).
> > 2. It seems that the Criteo dataset is a standard benchmark which is relevant to the studied problem (mentioned by another reviewer). The absence of experiments on it is a red flag. The authors state that it is too large to tune each model, but I don't find this convincing. The primary motivation for the method and use of FMs in general is precisely for computational feasibility with large datasets. As the authors themselves state, these methods must "rank thousands of items in milliseconds, billions of times per day." Based on a quick Google search, the Criteo dataset has ~52 million rows. This is large, but it doesn't seem out of scope given the previous claims, especially if the authors have access to industrial compute resources.
> >
> > Overall, while I appreciate the improvements the authors have made, I am left with more concerns than I originally had.

---

### Official Review · Reviewer_c3Zp · 2023-11-09

**Soundness:** 2 fair
**Presentation:** 2 fair
**Contribution:** 2 fair
**Rating:** 5
**Confidence:** 3

**Summary:**

Considering the challenges of numerical feature modeling in FMs, the paper proposes to encoding numerical features into a vector of function values for a set of basis functions. Among them, the author suggested using B-Spline which has approximation power, and proved their effectiveness through numerical evaluation. Additionally, the authors show how the proposed strategy can be easily integrated into existing systems.

**Strengths:**

1. It is reasonable to use B-Spline for fitting.
2. The proposed strategy can be integrated into existing systems with minor modifications.
3. The proposed spanning properties are interesting.
4. Validated on a real online system.

**Weaknesses:**

1. The author states that the proposed strategy can maintain training and inference efficiency, but no relevant experimental results have been verified.
2. Based on the existing findings (de Boor), the author gives the benefits of using the B-Spline basis. However, how to ensure that this error bound with good properties can be extended to the FM model with multiple feature interactions.
3. The used datasets are not common datasets in the field of FM (such as Avazu and Criteo), which makes the effectiveness of the proposed method seem unconvincing.
4. In the online system, CTR prediction error measures the accuracy of model prediction, while the ranking performance between advertisements needs to be measured by the AUC metric.
5. The author verified the effectiveness of the proposed strategy on FFM. But how to ensure the universality on other FMs? In addition, a comparison with SOTA FMs is needed.

**Questions:**

1. For numeric fields, what are the strengths and weaknesses of using scalar transform encoding and binning encoding?
2. How to choose the number of break-points for a specific FM, and it requires sensitivity analysis. In addition, the author states that the choice of break-points may vary between fields, so the complexity of the choice seems to reduce the practicality of the proposed method.
3. Typo: "spanning coefficient depend on" -> "spanning coefficient depends on", "FM variants.." -> "FM variants."

**Details Of Ethics Concerns:**

No ethical issues.

---

> ### Author Response · Authors · 2023-11-17
>
> We kindly ask the reviewer to read the general comment above. It answers question (1). Regarding question (2), it can indeed be complex if we want to win a Kaggle competition - take a dataset and apply it to all numerical fields by customizing the appropriate number of break points to every field. However:
> (1) we can see in the numerical experiments on public datasets, that even setting the number of bins to the same value for all numerical fields already provides a significant boost over binning.
> (2) in real recommender systems, fields are often added or changed incrementally, rather than all at once. So a data scientist working on feature engineering for a particular field can conduct the required experiments to choose the appropriate number of break points for their field.

---

> ### Author Response · Authors · 2023-11-19
> **Weakness 2 - AUC measurement.**
>
> It is important to understand that this is an online A/B test, rather than the evaluation on a data-set. In this case, metrics such as LogLoss or AUC are not informative at all. This is mainly because in advertising ads are **not** selected based on CTR, but based on the expected revenue $\mathrm{CTR} \cdot \mathrm{bid}$. The better we predict CTR, the better we predict the expected revenue, and the better our marketplace performs.  One model may cause more ads from a high-CTR category to win the auction, whereas its competitor may cause more ads with a lower CTR but higher bid ads to win auctions. Since the set of ads displayed to the user may be radically different between the two experimental environments, the "test sets" are different, metrics such as LogLoss or AUC are not comparable on real traffic.
>
> CTR prediction error, on the other hand, is much more informative. When the model over-predicts, it causes ads to unjustifiably win auctions, and due to the over-prediction, the true CTR will be lower than the predicted one. When the model under-predicts, it causes some ads to unjustifiably lose auctions, they will not be displayed to the user, and we will not observe these under-predictions at all. Thus, on average, models over-predict on real traffic that is displayed to the user, and the amount of over-prediction is an informative indicator of the model's accuracy.
>
> Of course, the best indicator is revenue. However, measuring revenue requires an experimentation environment where campaign budgets are not shared between the two experimental environments, otherwise the environment with the significantly less accurate model "steals" budgets from the environment with the significantly more accurate model. As is apparent in the paper, the prediction error before changing the field's representation was quite high. Such an environment has limited availability, and our team uses it only for very few high priority experiments. Thus, we were not able to report the revenue increase. Please see  [this](https://arxiv.org/abs/2012.08724) and [this](https://arxiv.org/abs/1605.09171) papers about the budget isolation issue.
>
> The above are not obvious for readers not experienced in online advertising, but we were not able to find a good reference explaining all of the above concerns in one place. We hope that this explanation helps the reviewer.

---

> > ### Comment · Reviewer_c3Zp · 2023-11-21
> > **Response to the Authors**
> >
> > Thank you for the thorough explanations. My main concern is the applicability of the proposed strategy to the state-of-the-art FMs. In addition, the authors emphasized in their response that they aim at systems where speed is of paramount importance, which further illustrates that large-scale datasets such as Criteo are a better choice.

---

### Official Review · Reviewer_xr1T · 2023-11-09

**Soundness:** 2 fair
**Presentation:** 1 poor
**Contribution:** 2 fair
**Rating:** 5
**Confidence:** 3

**Summary:**

The paper proposes a new method to map a given features onto a new feature space which is a combination of step functions. The paper provides an example of  a dataset with age, device type, and time the user spent on a website and states that "*for the segment of 25-years old users using an iPhone the model will learn some step function, whereas for the segment of 37-years old users using a laptop the model may learn a (possibly) different step function*". The proposed method tries to find the best mapping onto step functions which results in training a model with highest prediction ability. The paper states that this approach can play an important role in recommendation systems.

**Strengths:**

- This paper is about using factorization machines for encoding features and the proposed method tries to improve the accuracy of such methods. Although I am not familiar with factorization machines, the paper states that factorization machine are used effectively in recommender systems.
- The paper tested the performance of the proposed method on several real datasets and the paper examines the theoretical results on a synthetic dataset.

**Weaknesses:**

- I think there are significant shortcomings in the paper's clarity, presentation and organization. After reading the paper multiple times, I find it challenging to discern the paper's contributions and the study's importance. Specifically, section 2 is not written properly. Also the presentation of results in section 4 needs to be improved. It would be nice if authors lists the baselines and dataset description in a more organized manner for example at the beginning of section 4. Overall, I believe substantial revisions are needed to enhance the clarity and presentation of the paper.
- An important question that I could not find the answer in the paper is that why do we need to transform features into step function like features? Instead you can use neural networks as an example to produce a representation for each given feature vector and then the learned representation can be employed to train a model. This probably can be added as a baseline to the evaluation section.
- Based on my understanding, this paper focuses on improving factorization machine based methods and in section 4 the proposed method performance is only compared to factorization machine based models. I believe the paper does not motivate the importance of study well compared to simple methods such as representation learning with neural networks. This limits the contribution of this paper.

**Questions:**

Please see the second item in weaknesses section.

---

> ### Author Response · Authors · 2023-11-16
> **Reply**
>
> We kindly ask the reviewer to read the general comment above first. It addresses some of your concerns.
>
> Regarding the second weakness of using step functions. This is not our contribution, but a standard practice with FMs, since they can only represent a linear function of any given feature. Binning is a standard non-parametric technique that allows simple models  to approximately learn arbitrary function using a piecewise-constant representation. We also refer the reader to the paper "Practical Lessons from Predicting Clicks on Ads at Facebook" where additional, more advanced binning techniques are discussed in Section 3.1. Our contribution is an improvement over binning that is only slighly more complex, but significanlty more accurate.

---

### Official Review · Reviewer_Ema4 · 2023-11-10

**Soundness:** 2 fair
**Presentation:** 2 fair
**Contribution:** 2 fair
**Rating:** 5
**Confidence:** 2

**Summary:**

The paper establishes a connection between Factorization Machines (FMs) and approximators for segmentized functions, under the assumption that the remaining fields being constant. The authors propose to use B-Spline basis functions to represent numerical features within FMs. This approach is applicable to various FM variants, and the authors present a methodology for seamlessly integrating the encoding technique into existing systems through binning simulation. Two lemmas are presented to illustrate the relationship between the basis and FM output for individual and pairwise fields. Experiment results on simulation, public dataset and a/b testing demonstrate the effectiveness of the proposed method.

**Strengths:**

- The proposed approach is straightforward yet effective, adept at accurately approximating step functions with just a few knots, proving advantageous in scenarios with limited available samples.
- The method not only outperforms binning in simulated and public data scenarios but also significantly enhances the production model, showcasing considerable promise.
- The authors acknowledge certain limitations of the proposed method, such as its performance on product prices.

**Weaknesses:**

- The authors overlooked the exploration of deep learning-based architectures, such as deepFM, which are widely employed in industry, especially for Click-Through Rate (CTR) prediction. It would be intriguing to observe the impact of the proposed method when integrated with a deep learning component in the model.
- The authors solely employ binning as a baseline, neglecting the significance of using standardized numerical values as an important baseline.
- The utilization of hyperparameter search for small datasets by the authors may not be practical for real-world use cases with large datasets. A preferable approach would involve conducting ablation studies, especially on factors like step-size, to guide the selection of values for practical use cases.

**Questions:**

The acknowledgment of certain limitations in the proposed method is commendable. However, it would be more beneficial if the authors could provide a potential solution. For instance, insights on effectively identifying situations where the proposed method may falter and offering guidance on addressing such issues would enhance the practical utility of the paper.

---

> ### Author Response · Authors · 2023-11-17
>
> We kindly ask the reviewer to read the general comment above. It addresses some of the concerns pointed out in the weaknesses section.
>
> One point regarding hyperparameter search. In practice, hyperparameter search can be, and **is** done in practice. For example, please look at the following works:
> 1. Michal Aharon, Amit Kagian, and Oren Somekh. 2017. Adaptive Online Hyper-Parameters Tuning for Ad Event-Prediction Models. In Proceedings of the 26th International Conference on World Wide Web Companion (WWW '17 Companion). International World Wide Web Conferences Steering Committee, Republic and Canton of Geneva, CHE, 672–679. https://doi.org/10.1145/3041021.3054184
> 2. Parker-Holder, J., Nguyen, V. and Roberts, S.J., 2020. Provably efficient online hyperparameter optimization with population-based bandits. Advances in neural information processing systems, 33, pp.17200-17211.
>
> Moreover, one hyperparameter configuration can be better for bins, whereas another can be better for splines. Especially parameters like the step-size, that heavily depend on the variance of gradients, that varies greatly when changing the representation basis of choice. Thus, with one set of hyperparameters we could conclude that bins outperform splines, whereas with another one, splines outperform bins. We wanted to demonstrate that the best splines can do is better than the best that bins can do.

---

### Official Review · Reviewer_fV9X · 2023-11-10

**Soundness:** 3 good
**Presentation:** 2 fair
**Contribution:** 2 fair
**Rating:** 3
**Confidence:** 3

**Summary:**

This paper proposes a spline function approximation to be incorporated into applications of Factorization Machines where a feature takes continuous real values: instead of discretizing the feature and transforming it into a one hot encoding of the bin it belongs to, the encoding of the feature is a set of spline function values applied to it. This is similar to the kernel trick in  non-linear regression, except that this is done in the context of factorization machines, which by definition also involve second order terms. In Lemma 1, it is shown that this encoding results in the learned functions taking the form of spline functions of the numerical feature whenever all other features are fixed. In Lemma 2, it is shown that if the procedure is applied to two features instead, then the resulting functions (when fixing all other features) are arbitrary products of spline functions (essentially, ``low rank spline functions"). The authors also propose ``integrating" the model into existing factorization machines by discretizing the output into bins. Experiments are performed on some synthetic datasets which show that the model learns the ground truth functions better than the traditional discretization approach. On real data, an improvement is shown compared to the baseline on CTR prediction datasets, and the results of A/B testing on a real world proprietary platform are shown: the performance improvement was significant and sufficient to convince the company to adopt the paper's method

**Strengths:**

The idea of using basis functions (instead of a disctretization into bins) in the context of factorization machines is excellent.

The synthetic data experiments are encouraging (although I feel it doesn't quite "go all the way", what is attempted here is certainly a non trivial task).

Clearly, this is a model which has already been adopted in an industrial setting and is therefore of great interest to the community.

**Weaknesses:**

1. It is hard to believe that the authors can 100 percent claim the originality: by their own admission, some similar strategies have been employed in the more general context of tabular data [1] (the only novelty is its application to CTR prediction). No part of the ideas fails to transfer directly to the context studied here.

2. The paper doesn't have much content: "Lemmas" 1 and 2 are nearly completely trivial. There is nearly zero substance from the conceptual point of view, 90 percent of the value of the paper is in the experiments. Instead of simply showing what the learned functions look like as in Lemmas 1 and 2, it would be better if some analysis of the function class capacity of the model were performed more rigorously.

3. Section 3.3 and 3.4 are a little bit confusing. Instead of using vague sentences such as "we use a function which maps $f$ to a compact interval .... and ensures that no regions of the interval are starved by the dataset...concretely we use sklearn". The authors should consider writing a mathematically consistent definition.

4. Even from the experimental point of view, the synthetic data experiments don't really show comprehensive results: the results only vary the boundaries (not the number of splines) in Figure 4, and the results in Figure 3 are only for a given configuration of the number of splines and boundaries: in particular, despite the fact that splines are universal approximators (as the authors point out as well), no single experimental configuration allows the authors to recover a ground truth function nearly exactly.


This is more of an industry/problem-oriented result, getting this into the main *research track* of an extremely prestigious conference such as ICLR seems like a hard sell.

======================Minor typos=====================



Page 2: “an elementary lemmas”=> “an elementary lemma”

End of section 2: “equation equation 1” => “Equation (1)”


At the beginning of Section 3, again, : “equation equation 1” => “Equation (1)”
Same thing at the beginning of Lemma 1.  Similarly, at the beginning of the appendix, I think the authors mean “formula (1)”, note “formula equation (1)”


Also around the same place: “all the aforementioned family member” => “all the aforementioned family members”


Beginning of Section A.2 (proof of Lemma 1: w.l.o.g should be w.l.o.g. or wlog
(see also the same issue in section 3.3)

Beginning of Section 3.4: “Introduction” shouldn’t be capitalized

There is a period missing at the end of equation (4).

Bottom of page 14: “by following similar logic to 4=>  “by following similar logic to 4”

Bottom of page 8 “closely related FwFM” should be “closely related to FwFM”

In the Discussion section “.In particular, the splines…” should be “. In particular, the splines…”







=================References==================


[1] Paul Covington, Jay Adams, and Emre Sargin. Deep neural networks for youtube recommendations. Recsys 2016

**Questions:**

1. At the end of section 2, could also write $M_{f_i,f_j}=I_{f_i=f_j}$ instead of “$M=P_f^\top P_e$ where $P_f$ is a matrix which extracts the components of the field $f$”? It seems the statement is quite unnecessarily vague.

2. Could you explain Sections 3.3 and 3.4 a little bit more mathematically in your next pdf upload?

3. It seems like all the datasets are actually relatively traditional tabular data (not really specific to recommender systems or even CTR prediction), why did you not try to include datasets where some features include user ID such as MovieLens 25M or Douban? Since the method is not really new outside of its application to factorization machines, it feels like this is a must.

---

> ### Author Response · Authors · 2023-11-16
> **Answers**
>
> First, we kindly ask you to read the general comment above. Below are answers to your questions:
> 1. We made our description of the factorization machine family more explicit in the appendix of the revision, and refer to it in Section 2.1. We hope it clarifies exactly what $\mathbf{P}_f$ and $\mathbf{P}_e$ are.
> 2. As pointed out in the general comment, “feature engineering” is itself a non-rigorous process. But we did write it in more mathematical terms, and hope it's clearer.
> 3. As pointed out in the general comment - our method is useful when there are numerical features.

---

### Official Review · Reviewer_cfSS · 2023-11-16

**Soundness:** 2 fair
**Presentation:** 2 fair
**Contribution:** 2 fair
**Rating:** 3
**Confidence:** 4

**Summary:**

This paper examines the problem of discretizing features in the setting of factorization machine. The solution is based on spline functions and the authors claim that this solution addresses the sparsity issue.

**Strengths:**

It is in general a very carefully written paper that seems to address certain issues that practitioners care about. It was a smooth read.

**Weaknesses:**

The paper made a few claims that are inconsistent with my understanding of the area though.

1. Why discretize continuous variables: I have the impression that the community is moving the other way around. For example, lightgbm/catboost put significant effort to convert categorical features into numerical ones for the purpose of speeding up computation. The experiments also do not seem to confirm there is any compute-performance gain.
2. Novelty of spline: spline-based regressions/approximations also seems to be have been around for very long, and we should already have abundant techniques to understand its statistical/generalization properties (e.g., how choice of bins may impact training and test performance tradeoff). I fail to see any new contribution from this paper.
3. Why specifically factorization machine: I am also not sure why the authors specifically choose factorization machine as the authors seem to be addressing a generic feature representation problem (again, going back to my point 1 that I feel unsure whether the problem is “real”).
4. I am also feeling a bit unsettling about the citations: many very old results related to spline most relevant to this work, some relevant new results that dont appear in premier venues, and some recent neurips-tier results that are only briefly discussed. It will be helpful to better connect this work with recent development of ML as I suspect not many understand this click through rate business these days (I thought the standard approach is some neural nets with many features these days?).

**Questions:**

Could you please kindly answer my questions in the weakness section?

---

> ### Author Response · Authors · 2023-11-16
> **Answering questions**
>
> Please refer to the general comment above, since it addresses most of your questions, except for weakness (2). Regarding this weakness, there are plenty of examples when an old technique provides a novel result,  or interesting insight in a new context. For example, low-rank factorizations are very old, but factorization machines were novel, and LORA training in language models is also novel. We believe that the combination with splines with factorization machines provides, on the one hand, interesting insight (segmentized, or "personalized" spline functions), while also being useful in practice for large scale real-time recommender systems.

---

### Author Response · Authors · 2023-11-16
**General comment to reviewers and AC**

1. We would like to reiterate the first paragraph of our introduction. We aim at systems where speed is of paramount importance. Training must be fast to adapt to the ever-changing marketplace conditions [1,2,3,5] , and inference must be extremely fast to rank thousands of items in milliseconds, billions of times per day. Such systems are typically driven by FM variants with binned numerical features are used a lot throughout the industry.  Our aim is not a modeling approach that improves upon some baselines on some datasets, but is also _useful in this context_. Therefore, we have not overlooked NN, kernel, or decision forest baselines, but rather deem them out of scope. We would like to kindly ask the reviewers to evaluate the paper from this perspective. We do understand that some reviewers may believe that the scope of research aiming at such applications may be too narrow for ICLR.
2. We indeed haven’t tested the speed of our solution, but such a test depends heavily on the implementation. However, we point out that B-Spline and similar bases are ubiquitously used in computer graphics and computer aided design for real-time rendering and deformation of curves and surfaces, and in font rendering. Rendering this very OpenReview page requires evaluating a huge number of B-Spline functions. They are _extremely_ fast to evaluate, even on old hardware, and are “battle-tested” for decades in this sense.
3. It appears that there is some confusion between classical matrix factorization and factorization machines among the reviewers. Factorization machine variants do not aim to factor the user-item interaction matrix. Rather, they aim to factor the quadratic coefficients matrix $A_{i,j}$ of the linear + pairwise feature interaction model
$$
  f(x) = a + \langle b, x \rangle + \sum_{i \neq  j} A_{i,j} x_i x_j,
$$
See [4] for the exact form of these factorizations for the various variants.
FMs can be applied to any tabular data-set. However, they are typically used in recommender systems when users and/or items are modeled via their features, rather than their identity, to overcome sparsity and cold-start issues. We note that classical MF is a special case of an FM obtained when we have only user-id and item-id that are one-hot encoded, and concatenated to form x.  Make the paper more accessible to readers not coming from the relevant background, we explicitly wrote the formulae for the factorization machine variants in an Appendix and referred to it from Section 2.1.
4. Consequently, our method cannot improve anything for tasks without numerical features of users or items, such as the classical MovieLens datasets. The well-known Criteo Display Advertising Challenge data-set is a good fit, since it has plenty of such features, but is too large to train on with hyper-parameter tuning to neutralize their effect at a reasonable computational cost. In fact, this was the main reason for the choice of data-sets in our paper - they are tabular data-sets with numerical columns and are small enough to tune the model for each data-set.  Another widely used benchmark for FMs is the Avazu data-set, but it has no numerical features.
5. Several reviewers pointed out that Section 3.3 is not formal enough. They are correct, but this is the “feature engineering” part in our approach. It is informal, since feature engineering, in itself, is an informal process. Thus, it’s a recommendation to the readers, rather than a theoretical and rigorous framework. We made it written in a little bit more mathematical terms in our revision.
6. We clarified the difference between “fields” and “features” in the introduction’s second paragraph, since it appears to confuse some reviewers who do not come from the recommendation system background, and fixed typos pointed out by the reviewers.

**References**:

[1] Andreas Lommatzsch, Benjamin Kille, and Sahin Albayrak. *Incorporat-
ing context and trends in news recommender systems*. _In Proceedings of the international conference on web intelligence_.

[2] David Sculley, Gary Holt, Daniel Golovin, Eugene Davydov, Todd Phillips, Dietmar Ebner, Vinay Chaudhary, Michael Young, Jean-Francois Crespo, and Dan Dennison. *Hidden technical debt in machine learning systems*. In _Advances in neural information processing systems (2015)_.

[3] Yang Zhang, Fuli Feng, Chenxu Wang, Xiangnan He, Meng Wang, Yan Li, and Yongdong Zhang. *How to retrain recommender system? a sequential meta-learning method*. _In Proceedings of the 43rd International ACM SIGIR Conference on Research and Development in Information Retrieval_.

[4] Chen Almagor and Yedid Hoshen. 2022. *You Say Factorization Machine, I Say Neural Network - It’s All in the Activation*. _In Proceedings of the 16th ACM Conference on Recommender Systems (RecSys '22)_.

[5] A. Shtoff and Y. Koren. *Mitigating Divergence of Latent Factors via Dual Ascent for Low Latency Event Prediction Models*. _In 2021 IEEE International Conference on Big Data (Big Data)_.

---

> ### Author Response · Authors · 2023-11-16
> **The use of scalar transform**
>
> Several reviewers asked about a simple scalar transform of a feature. Since an FM model is linear in any single feature, assuming the rest are kept constant, the model will learn a family of 'segmentized' linear functions of that feature. If a feature is transformed using a scalar transform $T(z)$, then it will learn functions of the form $a T(z) + b$. In particular, $T$ itself may be linear, such as a standardization transform.
>
> We indeed have not conducted such experiments, but it is well known that splines of degree $d$ can represent any polynomial of degree $\leq d$, and in particular a linear polynomial. It is reasonable to assume that if the 'optimal' score is a linear function, the model will learn it.

---

### Meta-Review · Area_Chair_CroQ · 2023-12-12

**Metareview:**

This paper investigates the problem of discretizing features in the context of factorization machines. The proposed solution is based on spline functions, and the authors claim that it addresses the sparsity issue.

------------

Strengths: The paper is well-written and easy to understand. The proposed approach is efficient and can accurately approximate step functions with only a few knots, which is advantageous in situations with limited data. The method outperforms binning in simulated and public data scenarios, and it also significantly improves the production model.

------------

Weaknesses: Several reviewers inquired about why the proposed approach was not compared to neural networks. For example, Reviewer xr1T stated that "Instead, you can use neural networks as an example to generate a representation for each given feature vector, and then the learned representation can be used to train a model. This could be added as a baseline to the evaluation section." I agree that not comparing to neural networks is a flaw.

Furthermore, I am not persuaded by the authors' claim that "the time required to train and make inferences using neural networks is significantly greater than that required for factorization machines." In my opinion, a more direct comparison to neural networks would be more informative, as it would allow for a more direct assessment of the performance gap. Second, the authors emphasize "the time required to train and make inferences using neural networks," but they do not report the running time of the proposed approach (the authors admit that "Regarding computational efficiency, we have not conducted such studies").

Another flaw is that the absence of Criteo benchmarking is a concern, as noted by Reviewer cQDt and reviewer zL9t.

**Justification For Why Not Higher Score:**

See the weaknesses above.

**Justification For Why Not Lower Score:**

N/A

---

### Decision · Program_Chairs · 2024-01-16

Reject